# Genome-wide association study of body fat distribution identifies adiposity loci and sex-specific genetic effects

Mathias Rask-Andersen [1], Torgny Karlsson[1], Weronica E. Ek[1] & Åsa Johansson[1]

Body mass and body fat composition are of clinical interest due to their links to cardiovascular- and metabolic diseases. Fat stored in the trunk has been suggested to be more pathogenic compared to fat stored in other compartments. In this study, we perform genome-wide association studies (GWAS) for the proportion of body fat distributed to the arms, legs and trunk estimated from segmental bio-electrical impedance analysis (sBIA) for 362,499 individuals from the UK Biobank. 98 independent associations with body fat distribution are identified, 29 that have not previously been associated with anthropometric traits. A high degree of sex-heterogeneity is observed and the effects of 37 associated variants are stronger in females compared to males. Our findings also implicate that body fat distribution in females involves mesenchyme derived tissues and cell types, female endocrine tissues as well as extracellular matrix maintenance and remodeling.

[1] Department of Immunology, Genetics and Pathology, Science for Life Laboratory, Uppsala University, Box 256, 751 05 Uppsala, Sweden. Correspondence and requests for materials should be addressed to M.R.-A. (email: mathias.rask-andersen@igp.uu.se) or to Å.J. (email: asa.johansson@igp.uu.se)

Overweight (body mass index [BMI] > 25) and obesity (BMI > 30) have reached epidemic proportions globally[1]. Almost 40% of the world's population are now overweight[2] and 10.8% are obese[3]. Obesity is set to become the world's leading preventable risk factor for disease and early death due to the increased risks of developing type 2 diabetes, cardiovascular disease, and cancer[4]. The distribution of adipose tissue to discrete compartments within the human body is associated with differential risk for development of cardiovascular and metabolic disease[5]. Body fat distribution is also well known to differ between sexes. After puberty, women accumulate fat in the trunk and limbs to a proportionally greater extent compared to other parts of the body, while men accumulate a greater extent of fat in the trunk[6]. Accumulation of adipose tissue around the viscera, the internal organs of the body, is associated with increased risk of disease in both men and women[7]. In contrast, the preferential accumulation of adipose tissue in the lower extremities, i.e., the hips and legs, has been suggested as a factor contributing to the lower incidence of myocardial infarction and coronary death observed in women during middle age[8]. The differential distribution of body fat between sexes has been attributed to downstream effects of sex hormone secretion[5]. However, the biological mechanisms that underlie body fat distribution have not been fully elucidated.

BMI is commonly used as a proxy measurement of body adiposity in epidemiological studies and in clinical practice. However, BMI is unable to discriminate between adipose and lean mass, and between fat stored in different compartments of the body. Other proxies that better represent distribution of body fat have also been utilized, such as waist circumference (WC), hip circumference (HC), and the waist-to-hip ratio (WHR). Through genome-wide association studies (GWAS), researchers have identified hundreds of loci to be associated with proximal measurements of body mass and body fat distribution such as BMI[9], WHR[10,11] and hip and waist circumference[11]. Sex-stratified analyses have revealed sexual dimorphic effects at twenty WHR-associated loci and 19 of these loci displayed stronger effects in women[12]. Body fat mass has also been studied in GWAS by using bio-electrical impedance analysis (BIA) and dual energy X-ray absorptiometry (DXA)[13,14]. BIA measures the electrical impedance through the human body, which can be used to calculate an estimate of the total amount of adipose tissue. The gold standard method for measurements of body fat distribution is computed tomography (CT) or magnetic resonance imaging (MRI). However, these methods are costly. A GWAS has been performed for subcutaneous and visceral adiposity, measured with computed tomography scans, albeit in a relatively limited number of individuals ($N = 10,577$)[15].

Developments in BIA technology has now allowed for cost-efficient segmental body composition scans that estimate of the fat content of the trunk, arms and legs[16] (Fig. 1a). In this study, we use segmental BIA (sBIA) data on 362,499 participants of the UK Biobank to study the genetic determinants of body fat distribution to the trunk, arms, and legs. For this purpose, we

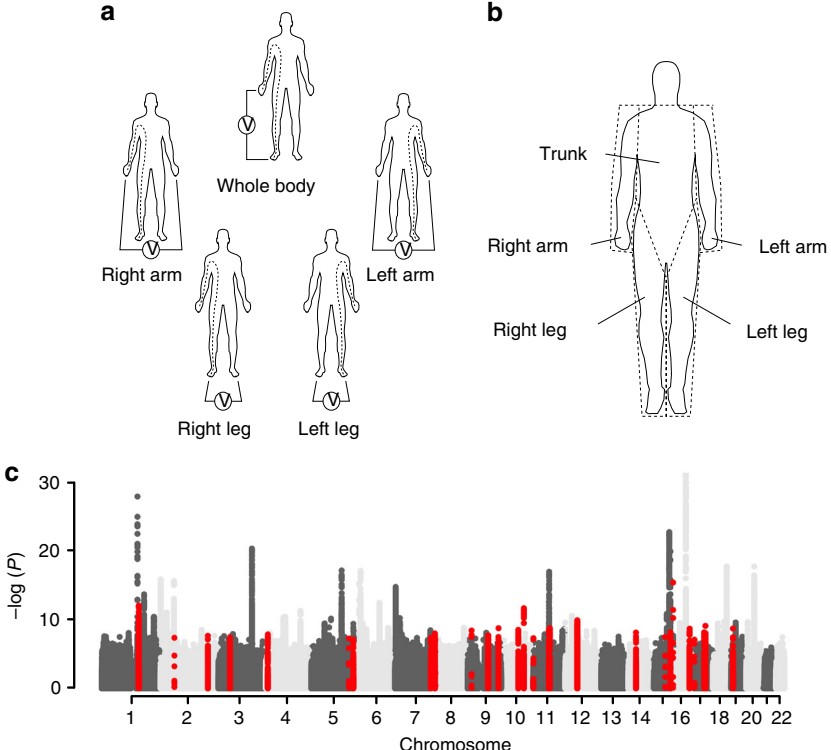

**Fig. 1** Segmental body impedance analyses. This method uses bio-electrical impedance to estimate body composition: fat mass, muscle mass, etc. In this study, adipose tissue mass was estimated using the Tanita BC-418MA body composition analyzer (**a**). This machine uses an eight-electrode method, which allows for five measurements of impedance. Electrical current is supplied to the front of both feet and the fingertips of both hands. Voltage is measured on either heel or thenar portion of the palms. Body composition is derived from a regression formula for each body part. The formula is derived from regression analysis using height, weight, age and impedance for each body part as predictors for composition of each body part as assessed by DXA (**b**). GWAS for AFR, LFR, and TFR were conducted in the UK Biobank cohort and revealed associations with loci that have not previously been associated with standard anthropometric traits. **c** A manhattan plot with combined results for association studies of body fat ratios in combined and sex-stratified analyses. Overall, 135 independent associations with at least one of the body fat ratios were observed in the discovery analyses. Out of the initial 135 associations, 98 replicated; of which 30 replicated for AFR, 44 for LFR and 66 for TFR. Loci that have not previously been associated with an anthropomorphic trait are highlighted in red ($N = 29$)

perform GWAS on the proportion of body fat distributed to these compartments. We also perform sex-stratified analyses to identify effects that differ between males and females.

We find 98 independent genetic signals to be associated with body fat distribution, as determined by sBIA, of which 29 have not previously been associated with any adiposity-related phenotype. We also find that genetic associations strongly differ between the sexes, in particular for distribution of adipose tissue to the legs and trunk where effects are primarily observed in females. Tissue enrichment analyses with DEPICT reveal mesenchyme-derived tissues, as well as tissues related to female reproduction to be important for distribution of adipose tissue to the legs and trunk in females.

## Results

### Genome-wide association studies for body fat ratios.
The proportions of body fat distributed to the arms—arm fat ratio (AFR), the legs—leg fat ratio (LFR), and the trunk—trunk fat ratio (TFR) were calculated by dividing the fat mass per compartment with the total body fat mass for each participant (Fig. 1a). We conducted a two-stage GWAS using data from the interim release of genotype data in UK Biobank as a discovery cohort. Another set of participants, for which genotype data were made available as part of the second release, was used for replication. After removing non-Caucasians, genetic outliers and related individuals, 116,138 and 246,360 participants remained in the discovery and replication cohorts, respectively. Basic characteristics of the discovery and replication cohorts are presented in supplementary Table 1. Females were found to have higher total sBIA-estimated fat mass compared to men in both the discovery and replication cohort. Males had higher average proportion of body fat located in the trunk compared to females (62.2% vs. 50.3%) and females had a larger proportion of body fat located in the legs (39.7% vs. 28.1%). While the total amount of adipose tissue in the arms was estimated to be higher in females compared to males, the fraction of adipose tissue distributed to the arms were similar. Several smaller differences between the discovery and replication cohorts were present (supplementary Table 1), such as some slight differences in height and age between men and women in the discovery and replication cohorts. These differences most likely represent the 50,000 participants for the UK Biobank Lung Exome Variant Evaluation (UK BiLEVE) project that were included in the first release of genotyping data for ~150,000 participants, which were used as a discovery cohort in this study. Selection for UK BiLEVE was conducted with specific consideration to lung function which may reflect the differences in baseline characteristics for this subset of the cohort. These participants were also genotyped on a separate but similar microarray and a batch variable was included in our association analyses to adjust for any effects related to the different genotyping arrays as well as having participated in the UK BiLEVE study (see methods).

GWAS was performed for each of the three phenotypes (AFR, LFR, and TFR) in the discovery cohort (sex-combined) and when stratifying by sex (males and females), while adjusting for covariates as described in the method section. A total of 25,472,837 imputed SNPs, with MAF of at least 0.0001, were analyzed in the discovery GWAS. LD score regression intercepts[17] ranged from 1.00 to 1.03 (supplementary Figs. 1–4, supplementary Table 2), and were used to adjust for genomic inflation. We used the --clump function in PLINK[18], in combination with conditioning on the most significant SNP, to identify associations that were independent within each GWAS as well as between the GWAS for the three body fat ratios (AFR, TFR, or LFR) or between strata (males, females or sex-combined;

see methods). For each independent association, the lead SNP, i.e., the SNP that was most significant in any of the phenotypes or strata, was taken forward for replication. In total, 135 independent associations were taken forward for replication of which 98 replicated (supplementary Tables 3–5, Supplementary Data 1). Substantial overlap in associated loci was observed between LFR and TFR loci (Fig. 2a) while AFR overlapped only to a small degree with LFR and TFR. One locus in the vicinity of ADAMTSL3 was associated with all three phenotypes.

**Overlap with findings from previous GWAS.** Body fat ratio-associated SNPs were tested for overlap with associations from previous GWAS for anthropometric traits by determining LD with entries from the GWAS-catalog[19]. In total, we identified 29 body fat ratio-associated signals that have not previously been associated with an anthropometric trait (Figure 1c, Table 1, supplementary Figs. 2–4). A large number of the body fat ratio-associated signals overlapped with previously identified height-associated loci[20,21] (supplementary Data 1) and the majority of these signals were associated with TFR and/or LFR (36 out of 40). For AFR, the strongest associations were observed at well known BMI and adiposity-associated loci such as: FTO, MC4R, TMEM18, SEC16B, and TFAP2B (supplementary Data 1).

We compared the direction of the effects for overlapping GWAS results by estimating the effects of lead body fat ratio-associated SNPs on the respective overlapping anthropometric traits. The effects of TFR-associated SNPs were directionally consistent with effects on height and WHR adjusted for BMI (WHRadjBMI), while the effects were the opposite for LFR. The direction of effects for AFR-associated SNPs were consistent with effects on BMI, WC, and WHR (supplementary Table 6).

Among the loci that have not previously been associated with an adiposity-related anthropometric trait, five overlapped with cardiovascular and metabolic trait-associated loci from previous GWAS: near XKR6, which is associated with carotid intima thickness[22], triglycerides[23,24], and systolic blood pressure[25,26]; ZNF652: coronary artery disease[27] and diastolic blood pressure[26,28]; RP11-32D16.1: diastolic- and systolic blood pressure[26,29]; RFTN: low HDL cholesterol[30]; and ERI1, which is associated with systolic blood pressure[26] (supplementary Table 7).

Sex-heterogenous effects of associated variants were tested for using the GWAMA software. This method utilizes summary statistics from sex-stratified GWAS to test for heterogeneity of allelic effects between males and females[31]. All replicated lead SNPs were included in these analyses. SNPs were only tested for heterogenous effects on the traits that they were associated with, which corresponds to 30 variants that were tested for sex-heterogenous effects on AFR, 44 on LFR and 66 on TFR. A striking heterogeneity in effects between males and females was observed (Table 2, supplementary Data 2). Two variants, near SLC12A2 and PLCE1, were shown to have larger effects on AFR in males while 37 variants exhibited larger effects in females. These variants were primarily associated with LFR and/or TFR (Table 2).

LD score regression (LDSC) was used to estimate the fraction of variance of body fat ratios that could be explained by SNPs, i.e., the SNP heritability[17]. SNP heritability was higher in females compared to males for all traits and ranged from ~21 to ~25% in females and from 11 to 15% in males (supplementary Table 2).

**Correlation between fat ratios and anthropometric traits.** Phenotypic and genotypic correlations were assessed, in males and females separately. Phenotypic correlations were estimated by calculating squared semi-partial correlation coefficients with ANOVA of nested linear models that were adjusted for age and

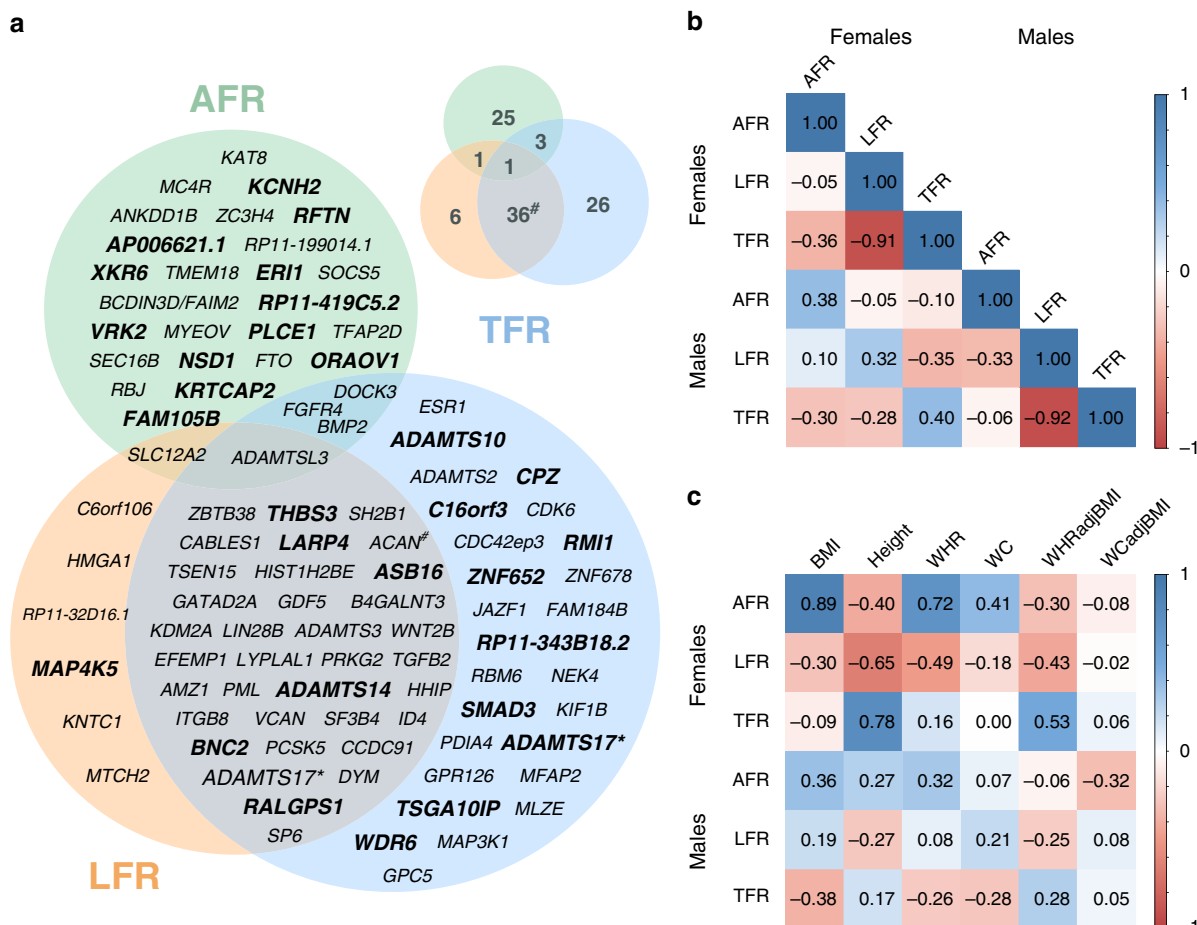

**Fig. 2** Overlap and genetic correlation between body fat ratios and other anthropometric traits. **a** The overlap between AFR-, LFR-, and TFR-associated loci is illustrated as a Venn diagram. The loci are denoted by the nearest gene or by the most likely target gene (see methods section). The loci in bold type and larger font designate loci that have not previously been associated with an anthropometric trait. The total number in each field is illustrated top right. *: Two independent signals were observed within the *ADAMTS17* locus: one with an effect on LFR and one on TFR. #: Two independent associations with LFR and TFR were observed within the *ACAN* locus. **b** Genetic correlation between body fat ratios within, and between sexes. Genetic correlations were estimated by cross-trait LD-score regression[32]. The absolute values for each genetic correlation ($r_g$) is included. **c** Genetic correlations between body fat ratios and standard anthropometric traits. Sex-stratified summary statistics were generated for each trait by GWAS in the discovery cohort. Color scales represent genetic correlation and range from red (−1.0: perfect anticorrelation) to blue (1.0 perfect correlation)

principal components while genetic correlations were estimated using cross-trait LD score regression[32] (see methods). Overall, the genetic and phenotypic correlations showed a large degree of similarity (supplementary Tables 8–9) and the correlations between the anthropometric traits and body fat ratios were directionally consistent for phenotypic and genetic correlations for all phenotypes. In females, BMI and WC was strongly correlated with AFR both with regards to phenotypic and genetic correlations (Fig. 2c, supplementary Tables 8–9). Height contributed to a moderate degree in explaining the phenotypic variance in LFR and TFR in females (16.0% and 25.3%, supplementary Table 8) despite the rather strong genotypic correlation between height and both LFR and TFR (Fig. 2c, supplementary Table 9). In males, anthropometric traits contributed only to a small degree in explaining the phenotypic variance of body fat ratios (supplementary Table 8). Consistent with this result, genetic correlations between body fat ratios and anthropometric traits in males were also quite low (Fig. 2c, supplementary Table 9).

Strong genotypic and phenotypic correlations were seen between LFR and TFR in both males and females with more than 82% of the variance explained (Fig. 2b, supplementary

Tables 8–9). LFR and TFR were inversely correlated, which agrees well with the large overlap in GWAS results for these phenotypes and the fact that the effect estimates from the GWAS was in the opposite direction for LFR and TFR (supplementary Data 1). In contrast, AFR appeared to be more independent, as only a low amount of phenotypic and genetic correlation was observed between AFR and LFR/TFR (Fig. 2b, supplementary Tables 8–9).

**Functional annotation of associated loci.** Functional annotation of the GWAS loci was performed by identifying overlap with eQTLs from the Genotype-Tissue Expression (GTEx) project[33] (supplementary Data 3) and by identifying potentially deleterious missense variants in LD ($R^2 > 0.8$) with our lead SNPs (supplementary Table 10). In total, 31 body fat ratio-associated loci overlapped with an eQTL, and 11 lead SNPs were in LD with a potentially deleterious missense variant. Polyphen and SIFT-scores were used to assess the deleteriousness of the variants. These scores represent the probability for functional effects of missense variants and were estimated through sequence analyses[34,35]. Missense variants were found in *ACAN*, *ADAMTS17*, *FGFR4* and *ADAMTS10*, where the lead SNPs were predicted to be damaging (supplementary Table 10). The

**Table 1 Body fat ratio-associated loci that have not previously been associated with an anthropometric trait**

| Chr | Lead SNP | Position (bp) | MAF | Most likely target gene | Strongest associated trait | $\beta_{Disc}$ | $P_{Disc}$ | $\beta_{Repl}$ | $P_{Repl}$ | $P_{Meta}$ | Direction |
|---|---|---|---|---|---|---|---|---|---|---|---|
| chr1 | rs4971091 | 155,143,768 | 0.377 | $KRTCAP2^\alpha$ | AFR - combined | −0.024 | $8.23*10^{-8}$ | −0.013 | $1.35*10^{-5}$ | $3.62*10^{-11}$ | −− |
| chr1 | rs180921974 | 155,268,131 | 0.023 | $THBS3^\delta$ | TFR - combined | −0.102 | $1.05*10^{-12}$ | −0.097 | $9.45*10^{-24}$ | $7.52*10^{-35}$ | −− |
| chr2 | rs62107261 | 422,144 | 0.047 | $FAM150B^\alpha$ | AFR - combined | −0.063 | $3.08*10^{-10}$ | −0.078 | $3.67*10^{-30}$ | $1.80*10^{-38}$ | −− |
| chr2 | rs13011472 | 57,961,602 | 0.489 | $VRK2^\alpha$ | AFR - combined | −0.024 | $4.81*10^{-8}$ | −0.013 | $6.96*10^{-6}$ | $1.17*10^{-11}$ | −− |
| chr2 | rs148812496 | 198,540,352 | 0.497 | $RFTN^\alpha$ | AFR - combined | −0.03 | $2.41*10^{-8}$ | −0.015 | $4.19*10^{-5}$ | $8.07*10^{-15}$ | −− |
| chr3 | rs4521268 | 49,137,904 | 0.333 | $WDR6^\beta$ | TFR - combined | −0.025 | $4.30*10^{-8}$ | −0.021 | $5.00*10^{-12}$ | $1.46*10^{-18}$ | −− |
| chr4 | rs2241069 | 8,602,798 | 0.461 | $CPZ^\alpha$ | TFR - females | 0.033 | $1.56*10^{-8}$ | 0.019 | $7.86*10^{-7}$ | $3.97*10^{-13}$ | ++ |
| chr5 | rs1317415 | 157,952,404 | 0.307 | $RP11-32D16.1^\alpha$ | LFR - females | −0.034 | $6.11*10^{-8}$ | −0.018 | $1.81*10^{-5}$ | $4.71*10^{-11}$ | −− |
| chr5 | rs34022431 | 176,677,563 | 0.026 | $NSD1^\alpha$ | AFR - combined | 0.071 | $8.55*10^{-8}$ | 0.04 | $7.12*10^{-6}$ | $1.67*10^{-11}$ | ++ |
| chr7 | rs56282717 | 150,657,095 | 0.243 | $KCNH2^\alpha$ | AFR - combined | −0.027 | $5.00*10^{-8}$ | −0.02 | $5.59*10^{-9}$ | $3.03*10^{-15}$ | −− |
| chr8 | rs2044387 | 8,907,950 | 0.425 | $ERI1^\alpha$ | AFR - females | 0.035 | $1.14*10^{-8}$ | 0.024 | $1.03*10^{-9}$ | $1.69*10^{-16}$ | ++ |
| chr8 | rs12546366 | 10,802,146 | 0.455 | $XKR6^\alpha$ | AFR - females | 0.031 | $9.29*10^{-8}$ | 0.021 | $6.50*10^{-8}$ | $8.23*10^{-14}$ | ++ |
| chr9 | rs10962638 | 16,846,111 | 0.142 | $BNC2^\alpha$ | TFR - combined | −0.037 | $4.22*10^{-9}$ | −0.014 | $3.90*10^{-4}$ | $4.50*10^{-10}$ | −− |
| chr9 | rs7039458 | 86,639,999 | 0.248 | $RMI1^{\beta, \delta}$ | TFR - females | 0.038 | $2.07*10^{-8}$ | 0.028 | $3.32*10^{-10}$ | $7.20*10^{-17}$ | ++ |
| chr9 | rs3780327 | 129,945,847 | 0.22 | $RALGPS1^\alpha$ | LFR - combined | −0.031 | $1.95*10^{-9}$ | −0.016 | $2.50*10^{-6}$ | $3.40*10^{-13}$ | −− |
| chr10 | rs34821335 | 72,433,203 | 0.273 | $ADAMTS14^\beta$ | TFR - females | 0.039 | $3.40*10^{-9}$ | 0.023 | $2.22*10^{-7}$ | $2.89*10^{-14}$ | ++ |
| chr10 | rs11289753 | 96,026,184 | 0.432 | $PLCE1^\alpha$ | AFR - combined | 0.03 | $2.41*10^{-12}$ | 0.021 | $1.60*10^{-12}$ | $1.22*10^{-22}$ | ++ |
| chr11 | rs1138714 | 825,110 | 0.432 | $AP006621.1^\alpha$ | AFR - combined | −0.024 | $5.47*10^{-8}$ | −0.012 | $2.50*10^{-5}$ | $6.65*10^{-11}$ | −− |
| chr11 | rs71455793 | 65,715,204 | 0.046 | $TSGA10IP^\delta$ | TFR - females | −0.082 | $4.45*10^{-9}$ | −0.039 | $1.85*10^{-5}$ | $8.29*10^{-12}$ | −− |
| chr11 | rs1789166 | 69,482,091 | 0.352 | $ORAOV1^\alpha$ | AFR - combined | −0.027 | $1.85*10^{-9}$ | −0.013 | $2.05*10^{-5}$ | $4.92*10^{-12}$ | −− |
| chr12 | rs11614785 | 50,880,422 | 0.341 | $LARP4^\alpha$ | TFR - females | 0.04 | $1.37*10^{-10}$ | 0.026 | $2.93*10^{-10}$ | $1.15*10^{-18}$ | ++ |
| chr14 | rs71420186 | 50,960,918 | 0.066 | $MAP4K5^\alpha$ | LFR - combined | −0.05 | $8.25*10^{-9}$ | −0.038 | $3.20*10^{-11}$ | $2.47*10^{-18}$ | −− |
| chr15 | rs35874463 | 67,457,698 | 0.059 | $SMAD3^\delta$ | TFR - females | 0.069 | $3.58*10^{-8}$ | 0.064 | $1.44*10^{-14}$ | $3.10*10^{-21}$ | ++ |
| chr15 | rs8026676 | 89,361,919 | 0.472 | $RP11-343B18.2^\beta$ | TFR - females | 0.034 | $7.24*10^{-9}$ | 0.02 | $2.71*10^{-7}$ | $6.37*10^{-14}$ | ++ |
| chr16 | rs8057620 | 69,884,619 | 0.46 | $RP11-419C5.2^\beta$ | AFR - females | 0.035 | $2.27*10^{-9}$ | 0.024 | $4.40*10^{-10}$ | $1.79*10^{-17}$ | ++ |
| chr16 | rs10584116 | 90,062,323 | 0.097 | $C16orf3^\beta$ | TFR - females | −0.052 | $8.99*10^{-8}$ | −0.033 | $6.97*10^{-7}$ | $1.20*10^{-12}$ | −− |
| chr17 | rs2071167 | 42,287,519 | 0.235 | $ASB16^\beta$ | LFR - combined | −0.028 | $1.84*10^{-8}$ | −0.021 | $1.05*10^{-6}$ | $1.58*10^{-10}$ | −− |
| chr17 | rs28394864 | 47,450,775 | 0.462 | $ZNF652^\alpha$ | TFR - females | −0.036 | $8.89*10^{-10}$ | −0.024 | $1.70*10^{-9}$ | $3.51*10^{-17}$ | −− |
| chr19 | rs62621197 | 8,670,147 | 0.029 | $ADAMTS10^\delta$ | TFR - females | −0.105 | $2.17*10^{-9}$ | −0.147 | $3.45*10^{-39}$ | $8.33*10^{-46}$ | −− |

Lead SNP denotes the strongest associated SNP at each locus. The Most likely target gene column denotes a gene related to the associated locus either by proximity to the lead SNP ($\alpha$), LD ($R^2 > 0.8$) with a lead eQTL SNP for the gene ($\beta$) or LD ($R^2 > 0.8$) with a missense variant within the gene ($\delta$). $\beta$—effect size estimate per allele. Meta analyses were performed with METAL[53]. Direction—summary of effect direction for the discovery and replication cohorts, with one + or − per cohort. The Disc and Repl subscripts denote effects and $P$-values in the discovery and replication cohorts, respectively. $P_{Meta}$ denotes the $P$-value for meta analyses

missense variant rs351855, within *FGFR4*, has also previously been shown to be associated with progression of cancer[36,37] and to affect insulin secretion in vitro[38].

**Enrichment analyses**. To identify the functional roles of body fat ratio-associated variants and which tissues are mediating the genetic effects, we performed enrichment analyses with DEPICT (Data-driven Expression Prioritized Integration for Complex Traits[39], see method section). In these analyses we used summary statistics from sex-stratified GWAS on the combined cohort (195,043 women and 167,408 men) in order to maximize statistical power. Results from the enrichment analyses were compared with results from previous GWAS for height, BMI[9] and WHRadjBMI[12]. Substantial overlap of enriched gene sets was observed between LFR/TFR with height and WHRadjBMI. In contrast, BMI-associated biological processes overlapped only to a marginal extent: 2% of all gene sets that were enriched for BMI-associated genes were also enriched for LFR- and TFR-associated genes (Fig. 3a).

Tissue enrichment was observed for LFR and TFR-associated genes in females (Fig. 3b, c) in gene sets related to female reproductive tissues, musculoskeletal tissues, chondrocytes, mesenchymal stem cells, and fibroblasts. For TFR, DEPICT also revealed enrichment of genes associated with adipose tissue cells, female urogenital organs, endocrine organs as well as the arteries (Fig. 3b). Tissue enrichment was not seen for the other traits or strata.

In the gene set analyses, enrichment was only detected for TFR- and LFR-associated genes in females as well as LFR-associated genes in males (supplementary Data 4). Gene sets related to bone morphology and skeletal development were among the most strongly associated with both LFR and TFR. We also find the TGFβ signaling pathway gene set to be enriched for genes within the TFR and LFR-associated loci in females, as well as SMAD1-, SMAD2-, SMAD3- and SMAD7 protein-protein

interaction subnetworks (supplementary Data 4), which act as TGFβ downstream mediators. There was a substantial overlap of enriched gene sets between TFR and LFR in females as well as moderate overlap with LFR-associated gene sets in males (supplementary Fig. 5). The large fraction of overlapping gene sets between LFR and TFR in females agrees well with the large overlap in GWAS signals.

## Discussion

In this study, we performed GWAS on distribution of body fat to different compartments of the human body and identified and replicated 98 independent associations of which 29 have not previously been associated with any adiposity-related phenotype. In contrast to earlier studies, we have not addressed the total amount of fat but rather the fraction of the total body fat mass that is located in the arms, legs and trunk. Body fat distribution is well known to differ between males and females, which we also clearly show in our study. We also show that the genetic effects that influence fat distribution are stronger in females compared to males. These results are consistent with previous GWAS that have revealed sexual dimorphisms in genetic loci for adiposity-related phenotypes, such as waist circumference and waist-to-hip ratio[10,40,41]. Phenotypic and genetic correlations, as well as results from GWAS and subsequent enrichment analyses, also revealed that the amount of fat stored in the arms in females is highly correlated with BMI and WC. This suggests that the proportion of fat stored in the arms will generally increase with increased accumulation of body mass and adipose tissue. In contrast, males exhibited moderate-to-weak phenotypic and genetic correlations between the distributions of fat to different parts of the body and anthropometric traits, which indicates that the proportions of body fat mass in different compartments of the male body remains more stable as body mass and body adiposity increases. Among the three phenotypes analyzed in this study LFR and TFR were inversely correlated in both males and

**Table 2 Sex heterogeneous effects of body fat ratio-associated SNPs was assessed with GWAMA[56] for all replicated trait-associated SNPs**

| SNP | Locus | Effect stronger in... | $P_{AFR}$ | $P_{LFR}$ | $P_{TFR}$ |
|---|---|---|---|---|---|
| rs9853018 | ZBTB38[γ] | females | | $1.28*10^{-16}$ | $2.29*10^{-26}$ |
| rs11856122 | ADAMTSL3[γ] | females | | $1.07*10^{-18}$ | $2.44*10^{-19}$ |
| rs28584580 | ACAN[γ] | females | | $2.96*10^{-11}$ | $7.40*10^{-15}$ |
| rs7680661 | HHIP[γ] | females | | $5.36*10^{-10}$ | $8.59*10^{-14}$ |
| rs41271299 | ID4[γ] | females | | $8.08*10^{-10}$ | $3.21*10^{-13}$ |
| rs9358913 | HIST1H2BE[γ] | females | | $1.22*10^{-13}$ | $1.28*10^{-12}$ |
| rs62621197 | ADAMTS10[δ] | females | | | $3.52*10^{-11}$ |
| rs3791679 | EFEMP1[γ] | females | | $5.74*10^{-7}$ | $4.39*10^{-11}$ |
| rs143384 | GDF5[γ] | females | | $7.20*10^{-6}$ | $4.43*10^{-11}$ |
| rs72755233 | ADAMTS17[γ] | females | | $4.18*10^{-8}$ | $8.94*10^{-11}$ |
| rs4800148 | RBBP8, CABLES1, C18orf45[γ] | females | | $3.78*10^{-10}$ | $1.15*10^{-10}$ |
| rs2145270 | BMP2[γ] | females | $2.88*10^{-4}$ | | $1.90*10^{-9}$ |
| rs3817428 | ACAN[γ] | females | | $5.97*10^{-10}$ | $3.01*10^{-9}$ |
| rs35344761 | PCSK5[γ] | females | | $3.06*10^{-8}$ | $5.66*10^{-9}$ |
| rs552846225 | KDM2A[γ] | females | | $3.66*10^{-6}$ | $1.71*10^{-8}$ |
| rs2273368 | WNT2B[γ] | females | | $1.58*10^{-7}$ | $1.83*10^{-8}$ |
| rs5779197 | TSEN15, GLT25D2[γ] | females | | $8.77*10^{-8}$ | $2.49*10^{-8}$ |
| rs11205303 | SF3B4[γ] | females | | $1.61*10^{-5}$ | $3.85*10^{-8}$ |
| rs10916174 | ZNF678[γ] | females | | | $6.29*10^{-8}$ |
| rs798491 | AMZ1[γ] | females | | $2.91*10^{-4}$ | $4.11*10^{-7}$ |
| rs481806 | JAZF1[γ] | females | | | $2.48*10^{-6}$ |
| rs11614785 | LARP4[γ] | females | | $6.28*10^{-5}$ | $4.01*10^{-6}$ |
| rs994014 | PRKG2[γ] | females | | $1.05*10^{-4}$ | $8.28*10^{-6}$ |
| rs2820443 | LYPLAL1[γ] | females | | $3.01*10^{-6}$ | $2.11*10^{-5}$ |
| rs991967 | TGFB2[γ] | females | | $1.82*10^{-4}$ | $5.62*10^{-5}$ |
| rs6570507 | GPR126[γ] | females | | | $6.11*10^{-5}$ |
| rs527582137 | CDK6[γ] | females | | | $1.60*10^{-4}$ |
| rs3823974 | ITGB8[γ] | females | | | $1.73*10^{-4}$ |
| rs12905253 | PML[γ] | females | | | $1.95*10^{-4}$ |
| rs115912456 | VCAN[γ] | females | | $4.59*10^{-5}$ | $2.22*10^{-4}$ |
| rs4988781 | ADAMTS17[γ] | females | | | $3.12*10^{-4}$ |
| rs55750792 | MFAP2[γ] | females | | | $3.14*10^{-4}$ |
| rs2492863 | C6orf106[γ] | females | | $3.80*10^{-8}$ | |
| rs9469762 | HMGA1[γ] | females | | $3.67*10^{-6}$ | |
| rs754537 | RBJ, DNAJC27[γ] | females | $1.58*10^{-14}$ | | |
| rs3812049 | SLC12A2[γ] | males | $2.90*10^{-7}$ | | |
| rs55872725 | FTO[γ] | females | $6.97*10^{-6}$ | | |
| rs11289753 | PLCE1[β] | males | $6.95*10^{-5}$ | | |
| rs8050894 | KAT8[γ] | females | $3.39*10^{-4}$ | | |

30 variants were tested for sex-heterogenous effects on AFR, 44 on LFR and 66 on TFR. P-values denote the results from tests for heterogeneity between sexes. Bonferroni correction was used to correct for multiple testing and P-values < 0.05/140 ($3.57*10^{-4}$) were considered significant. The Locus column denotes a gene related to the associated locus either in annotation from previous GWAS ([γ]), the most proximal gene ([β]) or LD ($R^2 > 0.8$) with a missense variant within the gene ([δ]). All gene names are in intalic font

females. This suggests that LFR and TFR to a large extent describe one trait, i.e., the distribution of adipose tissue between these two compartments, which is further supported by the large overlap in GWAS loci between the two phenotypes. In contrast, AFR was only weakly correlated with the other two traits.

Tissue enrichment revealed an important role in body fat distribution in females for mesenchyme-derived tissues: i.e., adipose and musculoskeletal tissues; as well as tissues related to female reproduction. This suggests that the distribution of fat to the legs and trunk in females is mainly driven by the effects of female gonadal hormones on mesenchymal progenitors of musculoskeletal and adipose tissues. Enrichment analyses also showed that LFR and TFR have unique features that separates them from other anthropometric measurements, which was indicated by the portion of LFR/TFR-associated gene sets that did not overlap with height-, BMI-, or WHRadjBMI-associated gene sets. However, there was also an overlap in the functional aspects between these traits with both height and WHRadjBMI. This is indicated by the tissue enrichment profile for LFR/TFR-associated genes, which shares features with tissue enrichments reported for height in previous GWAS[20] where height-associated genes were strongly enriched in musculoskeletal tissue types with additional enrichment in cardiovascular and endocrine tissue types, while. WHRadjBMI-associated genes[12] were enriched in adipocytes and adipose tissue subtypes. Of particular note, we did not identify any enrichment of body fat ratio-associated genes in CNS tissue gene sets in contrasts to enrichment analyses in previous GWAS for BMI where the CNS has been implicated in playing prominent role in obesity susceptibility[9].

In the GWAS for LFR and TFR in females, we find that several genes that highlight the influence of biological processes related to the interaction between cells and the extracellular matrix (ECM), as well as ECM-maintenance and remodeling. These include ADAMTS2, ADAMTS3, ADAMTS10, ADAMTS14, and ADAMTS17, which encode extracellular proteases that are involved in enzymatic remodeling of the ECM. Two lead SNPs were in LD ($R^2 > 0.8$) with potentially damaging missense mutations in ADAMTS10 and ADAMTS17 and two other GWAS signals overlapped with eQTLs for ADAMTS14 and ADAMTS3. In addition, possibly deleterious missense mutations in LD with our lead GWAS

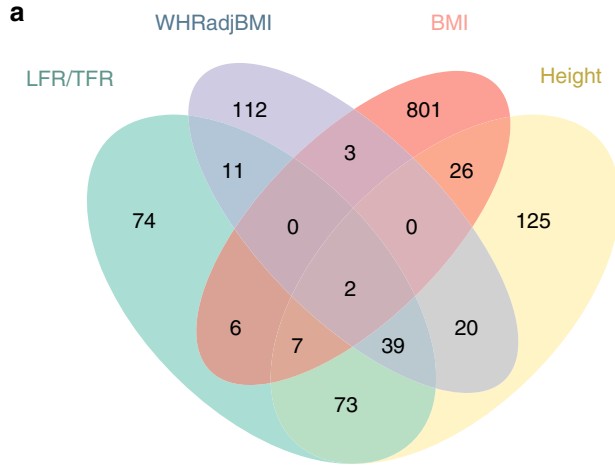

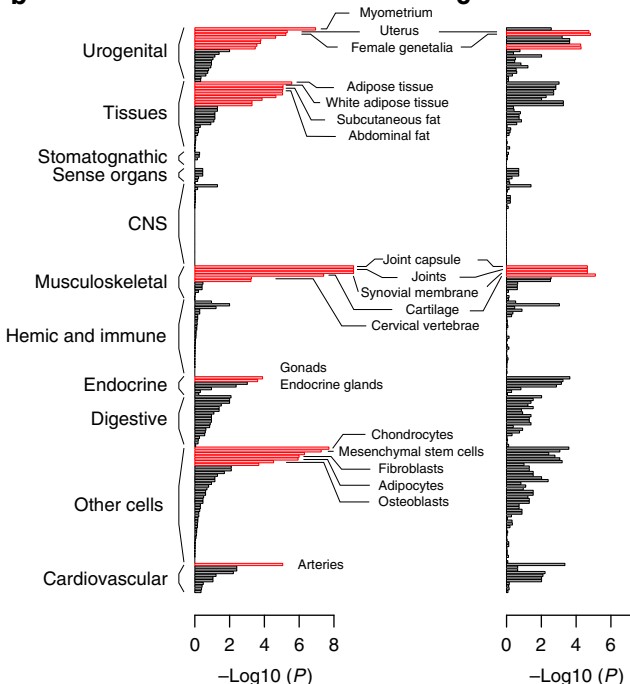

**Fig. 3** Enrichment analyses of genes at LFR and TFR-associated loci. **a** Reconstituted gene-sets that were enriched for TFR- and LFR-associated genes (in both males and females) were compared to results from previous GWAS on WHRadjBMI[12], BMI[9], and height[20]. Tissue and cell type enrichment of **b** TFR- and **c** LFR-associated genes in females. Red bars denote tissues gene sets that were significantly enriched for LFR- and TFR-associated genes at FDR < 0.05/12

SNPs were also found for *VCAN* and *ACAN*. Both *VCAN* and *ACAN* encode chondroitin sulfate proteoglycan core proteins that constitute structural components of the extracellular matrix, particularly in soft tissues[42]. These proteins also serve as major substrates for ADAMTS proteinases[43]. ECM forms the three-dimensional support structure for connective and soft tissue. In fat tissue, the ECM regulates adipocyte expansion and proliferation[44]. Remodeling of the ECM is required to allow for adipose tissue growth and this is achieved through enzymatic processing of extracellular molecules such as proteoglycans, collagen and hyaluronic acid. For example, the ADAMTS2-, 3-, and 14-proteins act as procollagen N-propeptidases that mediate the maturation of triple helical collagen fibrils[45,46]. We therefore propose that the effects of genetic variation in biological systems involved in ECM-

remodeling is a factor underlying normal variation in female body fat distribution.

In summary, GWAS of body fat distribution determined by sBIA reveals a genetic architecture that influences distribution of adipose tissue to the arms, legs, and trunk. Genetic associations and effects clearly differ between sexes, in particular for distribution of adipose tissue to the legs and trunk. The distribution of body fat in women has previously been suggested as a causal factor leading to lower risk of cardiovascular and metabolic disease, as well as cardiovascular mortality for women in middle age[5] and genetic studies have identified SNPs that are associated with a favorable body fat distribution[47], i.e., with higher BMI but lower risk of cardiovascular and metabolic disease. The capacity for peripheral adipose storage has been highlighted as one of the components underlying this phenomenon[47]. Resolving the genetic determinants and mechanisms that lead to a favorable distribution of body fat may help in risk assessment and in identifying novel venues for intervention to prevent or treat obesity-related disease.

## Methods

**UK Biobank participants**. The first release of imputed genotype data from UK Biobank ($N = 152,249$) was used as a discovery cohort, and an unrelated set of participants from the second release ($N = 326,565$) as a replication cohort. Imputed genotype data from the third UK Biobank genotype data release were used for replication. Participants who self-reported as being of British descent (data field 21000) and were classified as Caucasian by principal component analysis (data field 22006) were included in the analysis. Genetic relatedness pairing was provided by the UK Biobank (Data field 22011). Participants were removed due to relatedness based on kinship data (estimated genetic relationship > 0.044), poor genotyping call rate (<95%), high heterozygosity (Data field 22010), or sex-errors (Data field 22001). After filtering, 116,138 participants remained in the discovery cohort and 246,361 in the replication cohort.

**Ethics**. Ethical approval to collect participant data was given by the North West Multicentre Research Ethics Committee, the National Information Governance Board for Health & Social Care, and the Community Health Index Advisory Group. UK Biobank possesses a generic Research Tissue Bank approval granted by the National Research Ethics Service (http://www.hra.nhs.uk/), which lets applicants conduct research on UK Biobank data without obtaining separate ethical approvals. All participants provided signed consent to participate in UK Biobank[48].

**Genotyping, imputations, and QC**. Genotyping in the discovery cohort had been performed on two custom-designed microarrays: referred to as UK BiLEVE and Axiom arrays, which genotyped 807,411 and 820,967 SNPs, respectively. Imputation had been performed using UK10K[49] and 1000 genomes phase 3[50] as reference panels. Prior to analysis, we filtered SNPs based on call rate (--geno 0.05), HWE ($P > 10^{-20}$, Fischer's exact test), MAF (--maf 0.0001), and imputation quality (Info > 0.3) resulting in 25,472,837 SNPs in the discovery cohort. The third release of data from the UK Biobank contained genotyped and imputed data for 488,366 participants (partly overlapping with the first release). For our replication analyses, we included an independent subset that did not overlap with the discovery cohort. Genotyping in this subset was performed exclusively on the UK Biobank Axiom Array. This dataset included 47,512,111 SNPs that were filtered based on HWE ($P < 10^{-20}$, Fischer's exact test), call rate > 95% (--geno 0.05), Info > 0.3, and MAF > 0.0001. All genomic positions are in reference to hg19/build 37.

**Phenotypic measurements**. The phenotypes used in this study derive from impedance measurements produced by the Tanita BC-418MA body composition analyzer. Participants were barefoot, wearing light indoor clothing, and measurements were taken with participants in the standing position. Height and weight were entered manually into the analyzer before measurement. The Tanita BC-418MA uses eight electrodes: two for each foot and two for each hand. This allows for five impedance measurements: whole body, right leg, left leg, right arm, and left arm (Fig. 1a). Body fat for the whole body and individual body parts had been calculated using a regression formula, that was derived from reference measurements of body composition by DXA (Fig. 1b) in Japanese and Western subjects. This formula uses weight, age, height, and impedance measurements[51] as input data. Arm, and leg fat masses were averaged over both limbs. Arm, leg, and trunk fat masses were then divided by the total body fat mass to obtain the ratios of fat mass for the arms, legs and trunk, i.e., what proportion of the total fat in the body is distributed to each of these compartments. These variables were analyzed in this study and were named: AFR, LFR, and TFR.

**Correlations between fat ratios and anthropometric traits**. Phenotypic correlations between fat distribution ratios and anthropometric traits were estimated by calculating squared semi-partial correlation coefficients for males and females separately, using anova.glm in R. Adipose tissue ratios (AFR, LFR or TFR) were set as the response variable. BMI, waist circumference, waist circumference adjusted for BMI, waist-to-hip ratio, height, and one of the other ratios were included as the last term in a linear model with age and principal components as covariates. The reduction in residual deviance, i.e., the reductions in the residual sum of squares as BMI, waist circumference, waist circumference adjusted for BMI, waist-to-hip ratio, height, or one of the other ratios was added to the model, is presented as percentages of the total deviance of the null model in supplementary Table 8.

**Genome-wide association studies for body fat ratios**. A two-stage GWAS was performed using a discovery and a replication cohort. Sex-stratified GWAS were performed in the discovery cohort for each trait. A flowchart that describes the steps taken for the genetic analyses is included as supplementary Fig. 6. Prior to running the GWAS, body fat ratios were adjusted for age, age squared and normalized by rank-transformation separately in males and females using the *rntransform* function included in the GenABEL library[52]. GWAS was performed in PLINK v1.90b3n[18] using linear regression models with the age-adjusted and rank-transformed AFR, LFR, and TFR as the response variables and the SNPs as explanatory variables. 50,000 participants of UK Biobank were genotyped on a separate array as part of the UK BiLEVE project. A batch variable was used as covariate in the GWAS for the discovery analyses to adjust for genotyping array (UKB Axiom and UK BiLEVE) as well as for other differences between UK BiLEVE and UKB Axiom-genotyped participants. We also included the first 15 principal components and sex (in the sex-combined analyses) as covariates in the GWAS. LD score regression intercepts (see further information below), calculated using ldsc[17], were used to adjust for genomic inflation, by dividing the square of the *t*-statistic for each tested SNP with the LD-score regression intercept for that GWAS, and then calculating new *P*-values based on the adjusted *t*-statistic. We used a threshold of $P < 10^{-7}$, after adjusting for LD score intercept, as a threshold for significance in the discovery cohort.

The --clump function in PLINK was used to identify the number of independent signals in each GWAS. This function groups associated SNPs based on the linkage disequilibrium (LD) pattern. The parameters for clumps were set to: --clump-p1 $1*10^{-7}$, --clump-p2 $1*10^{-7}$, --clump-r2 0.10, and --clump-kb 1000. This function groups SNPs within one million base pairs that were associated with the trait at $P < 1*10^{-7}$. After running --clump in PLINK, conditional analyses were also performed for each locus conditioning for the lead SNP, but no further signals were identified. Several associations were found in more than one of the three body fat ratios (AFR, TFR, or LFR) or strata (males, females, or sex-combined) and different lead SNPs were observed for different traits and strata at several loci. To assess whether these represented the same signal, we assessed the LD between overlapping lead SNPs in PLINK. SNPs in low LD ($R^2$-value < 0.05) were considered to represent independent signals. We then performed conditional analysis in PLINK, conditioning on the most significant SNPs across all phenotypes and strata. Lead SNPs with a $P < 1*10^{-7}$ after conditioning on other potentially linked $R^2$-value ≥ 0.05 lead SNPs, were considered as being independent signals. For each independent signal, the lead SNP (lowest *P*-value) was taken forward for replication. Bonferroni correction was used to correct for multiple testing during replication and p-values < 0.05/135 were considered to be statistically significant. Meta analyses of results from the discovery and replication cohorts was performed with the METAL software[53] for all independent associations that were taken forward for replication.

**SNP heritability and genetic correlations**. We estimated SNP heritability and genetic correlations using LD score regression (LDSC), implemented in the ldsc software package[17]. Only SNPs that were included in HapMap3 were included in these analyses. LDSC uses LD patterns and summary stats from GWAS as input. For genetic correlations, we performed additional sex-stratified GWAS in the UK biobank (using the same covariates as for the ratios) for standard anthropometric traits, BMI, height, WC, WHR, WCadjBMI, and WHRadjBMI, in the discovery cohort. GWAS summary stats were filtered for SNPs included in HapMap3 to reduce likelihood of bias induced by poor imputation quality. After this filtering, 1,164,192 SNPs remained for LDSC analyses. LD scores from the European data of the 1000 Genomes project (including LD patterns for all the HapMap3 SNPs) for use with LDSC were downloaded from the Broad institute at: https://data.broadinstitute.org/alkesgroup/LDSCORE/eur_w_ld_chr.tar.bz2. Genetic correlations between the three body fat ratios and anthropometric traits were assessed by cross-trait LD score regression.

**Overlap with findings from previous GWAS**. Lead SNPs from all independent signals in our analyses were cross-referenced with the NHGRI-EBI catalog of published genome-wide association studies (GWAS Catalog—data downloaded on 23 April 2018)[19] to determine whether body fat ratio-associated signals overlapped with previously identified anthropometric associations from previous GWAS. We used a cut-off of $R^2 < 0.1$ between SNPs from our analyses and anthropometric

trait-associated SNPs ($P < 5*10^{-8}$) from GWAS catalog to determine any overlap with findings from previous GWAS. LD between data in the GWAS catalog and our lead SNPs were calculated using PLINK v1.90b3n[18]. In addition, lead SNPs at Body fat ratio-associated loci that potentially overlapped ($R^2 > 0.1$) with signals from previous GWAS were tested for association with standard anthropometric traits (BMI, height, WC, WCadjBMI, and WHRadjBMI) in the UK biobank discovery cohort using PLINK v1.90b3n[18] through linear regression modeling and including sex, age a batch variable and 15 principal components as covariates. Here, a $P < 1e{-}7$ was considered significant.

**Functional annotation of associated loci**. Associated loci were investigated for overlap with eQTLs from the GTEx project[33]. The threshold for significance for the eQTLs was set to $2.3*10^{-9}$ in agreement with previous studies[54]. The strongest associated SNP for each tissue and gene in the GTEx dataset was identified. We then estimated the LD between the top eQTL SNPs and the lead SNP for each independent association from our analysis. If a SNPs from our analyses were in LD ($R^2 > 0.8$) with a lead eQTL SNP the two signals were considered overlapping.

Lead SNPs, and all SNPs in LD ($R^2 > 0.8$) with a lead SNP from our analyses (LD determined in the UK biobank cohort in PLINK) were cross-referenced with dbSNP (human 9606 b150) in order to identify potentially deleterious intragenic variants in LD ($R^2 > 0.8$) with the body fat ratio-associated variants. Polyphen and SIFT-scores for the missense variants (extracted from Ensembl—www.ensembl.org) were used to assess the deleteriousness of the body fat ratio-associated variants.

**Enrichment analysis**. To identify the functional roles and tissue specificity of associated variants, we performed tissue and gene-set enrichment analyses using DEPICT[39]. For the gene-set enrichment in DEPICT, gene expression data from 77,840 samples have been used to predict gene function for all genes in the genome based on similarities in gene expression. In comparison to standard enrichment tools that apply a binary definition to define membership in a set of genes that have been associated with a biological pathway or functional category (genes are either included or not included), in DEPICT, the probability of a gene being a member of a gene set has instead been estimated based on correlation in gene expression. This membership probability to each gene set has been estimated for all genes in the human genome and the membership probabilities for each gene have been designated reconstituted gene sets. A total of 14,461 reconstituted gene sets have been generated which represent a wide set of biological annotations (Gene Ontology [GO], KEGG, REACTOME, Mammalian Phenotype [MP], etc.). For tissue enrichment in DEPICT, microarray data from 37,427 human tissues have been used to identify genes with high expression in different cells and tissues.

For the enrichment analyses, we performed sex-stratified GWAS for AFR, LFR and TFR on the combined cohort, i.e., the discovery and replication cohorts, including 195,043 females and 167,408 males, in order to achieve higher power. The clump functionality in PLINK is used to determine associated loci. The *P*-value cut-off for clump was set at $P < 10^{-7}$. In the enrichment analyses, DEPICT assesses whether the reconstituted gene sets are enriched for genes within trait-associated loci[39]. The false discovery rate (FDR)[55] was used to adjust for multiple testing. Twelve analyses were run in total (tissue enrichment and gene-set enrichment) and FDR < 0.05/12 was considered significant.

**Interaction between SNPs and sex**. We used the GWAMA software[31] to test for heterogenous effects of associated SNPs between sexes. In GWAMA, fixed-effect estimates of sex-specific and sex-combined beta coefficients and standard errors are calculated from GWAS summary statistics to test for heterogeneous allelic effects between females and males. GWAMA obtains a test-statistic by subtracting the sex-combined squared *t*-statistic from the sum of the two sex-specific squared *t*-statistics. This test statistic is asymptotically $\chi^2$-distributed and equivalent to a normal *z*-test of the difference in allelic effects between sexes. Lead SNPs that replicated were tested for heterogeneity between sexes for the trait that they were associated with. This corresponds to 30 tests for AFR, 44 for LFR, and 66 for TFR. Bonferroni correction was used to correct for multiple testing and $P$-values $< 0.05/140 = 3.57*10^{-4}$ ($\chi^2$ test) were considered to be significant. Summary statistics from the replication cohort were used in order to maximize statistical power.

## Data availability
The data that support the findings of this study are available from UK Biobank (http://www.ukbiobank.ac.uk/about-biobank-uk/). Restrictions apply to the availability of these data, which were used under license for the current study (Project No. 15152). Data are available for bona fide researchers upon application to the UK Biobank. Summary statistics from all association tests are available for download at: https://myfiles.uu.se/ssf/s/readFile/share/3993/1270878243748486898/publicLink/GWAS_summary_stats_ratios.zip. All other relevant data are available from the authors.

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

## Acknowledgements

We are grateful to the participants and staff of the UK Biobank. Access to UK Biobank genetic and phenotypic data were granted under application no. 15152. Computations were performed on the computational cluster at the Uppsala Multidisciplinary Center for Advanced Computational Science (UPPMAX) under projects b2016021, b2017066, and sens2017538. The work was supported by grants from the Swedish Society for Medical Research (SSMF), the Kjell and Märta Beijers Foundation, Göran Gustafssons Foundation, the Swedish Medical Research Council (Project Number 2015-03327), the Marcus Borgström Foundation, The Swedish Heart-Lung foundation, and the Åke Wiberg Foundation.

## Author contributions

M.R.-A., T.K., W.E.E., and Å.J. conceived of and designed the study. Analysis was performed by M.R.-A. and W.E.E. under supervision by Å.J., M.R.-A. analyzed the data and

wrote the first draft of the manuscript. All authors contributed to the final version of the manuscript.

## Additional information

**Competing interests:** The authors declare no competing interests.

