## [Peer Review File · Nature Communications]

Reviewer #1 (Remarks to the Author):

The authors have made their effort to address the reviewers comments to the highest extent. Overall the manuscript improved greatly and the discussion reads well. Results are still a bit of enumerating style and could be improved. I would be curious to discover, whether authors would provide any insights about AFR, which shows highest correlation with BMI, being a better proxy to detect metabolically unhealthy adiposity, as suggested by some authors.

Major comments

Part Overlap with findings from previous GWAS is out of place – it would fit better at the beginning of results, when authors state the totals (that part is hard to follow as it is full of total counts, which still don't provide a great deal of novelty of information. The total counts might be provided in F1 together with phenotype definitions).

Sex differentiated effects table is not clear – the effects in males or females columns can be abolished, and the sex with larger effect can be mentioned before each trait-specific p-value. Justification for Multiple testing correction for 97 tests is not sufficient, unless authors state the high correlation between the three studies measures. The legend doesn't make the table content clearer.

Line 135 – would suggest mentioning the six loci associated in males combined with the phenotype associated. The authors could speak about sex-specific effects, if association was not significant in one sex. Current terminology is pretty weak.

Genetic correlation section would benefit from a figure, which would nicely fit as Figure b in current F2.

Most of the DEPICT results section is enumerating the totals rather than providing biological insights – this part could be improved.

Minor points:

The starting phrase of the main text doesn't read well.

The phrase "has been shown to be associated with" is heavy and says the same as "is associated".

Line 99 is repetitive to line 97.

Reviewer #2 (Remarks to the Author):

The manuscript has been substantially shortened and clearer with respect to the methods and presentation of results. There is also effort in providing a global view using enrichment analyses and comparison with other standard anthropometric traits.

- Tables 1 and 2 are a little disjoint (or I am not reading it correctly). Was GWAMA used on the set of variants taken forth for replication in sex-combined results or the set of 6+71 sex-specific variants that replicated? It is still not clear to me how the sex-heterogenous effects were assessed.

- Did the authors do a full meta-analysis of the two sets of data? If yes, how did the results differ from the staged approach? I am not fully convinced that a staged approach is more robust and it is messier to assess the different signals at the same locus for males, females. May I suggest that the authors consider a simple flowchart to illustrate all the steps that they have taken in their analyses?

- The authors mentioned that the baseline difference in characteristic of 50,000 individuals in the discovery cohort was unlikely to affect the results from the analyses. Was anything done to assess that? Including study type as covariate or any sensitivity analysis? 50,000 is a third of the

discovery.

- Exact conditional analyses should be used for assessing independence of signals when individual-level data are available. LD is only an approximation and should be used as supporting evidence.
- It is a little of a stretch to say that missense variants at the loci can be "plausible causal".
- "lead SNP" is more commonly used than "leading SNP".

Reviewer #3 (Remarks to the Author):

The authors addressed many of the issues raised in the previous review of the manuscript. However, issues remain or developed in the revision.

1. A primary claim of the paper is that these analyses identified loci (Table 1) that have not previously been associated with an anthropometric trait, based on LD $r^2 < .1$ with loci listed in the NHGRI-EBI catalog. However, several loci that I checked are not novel using these or even more rigorous criteria. For example, line 203 results text highlights two 'novel' loci, KAT8 and ANKDD1B, but at KAT8, rs8050894 reported here is in $r^2 = .86$ (1000Genomes phase 3 EUR) with rs9925964 reported previously as associated with BMI by Locke et al in 2015; and at ANKDD1B, rs34341 reported here is in $r^2 = .54$ with rs6881648 reported as associated with BMI by Akiyama et al in 2017. Further, on line 205, the rs2145270 SNP at highlighted locus BMP2 was itself described previously as associated with BMI by Willer et al 2013. Also, rs2492863 reported here is in $r^2 = .99$ with rs2744971 reported as associated with height by Berndt et al in 2013. Other loci may also overlap previous reports. Given the claim that novel biology can be detected by novel loci detected using these AFR, LFR, and TFR traits, the loci need to be reviewed again more rigorously to ensure that they meet the authors' definition of novel. The loci that are truly novel for any anthropometric trait would be interesting and could help explain the basis for differences in body fat distribution.

2. Based on the further review of loci that are novel for a trait or overlap between traits, Figure 2 likely needs to be updated. The loci that are uniquely associated with the AFR, LFR or TFR traits, not shared by any of the other anthropometric traits, would provide the most novel and interesting information.

3. The Table 3 missense variants seem to be mostly from previously reported anthropometric trait loci and do not provide substantial new information. This Table could be moved to the Supplement.

Reviewers' comments:

Reviewer #1 (Remarks to the Author):

The authors have made their effort to address the reviewers' comments to the highest extent. Overall the manuscript improved greatly and the discussion reads well. Results are still a bit of enumerating style and could be improved.

Authors' response: We greatly appreciate the supportive comments from Reviewer #1. We have adjusted the results section to get away from the enumerating style that was pointed out by Reviewer #1.

Major comments

1. Part Overlap with findings from previous GWAS is out of place – it would fit better at the beginning of results, when authors state the totals (that part is hard to follow as it is full of total counts, which still don't provide a great deal of novelty of information. The total counts might be provided in F1 together with phenotype definitions).

Authors' response: We have moved the part of the manuscript that concerns overlap with results from previous GWAS to directly follow the section with the main results from the GWAS for the different traits. In addition, we have moved the total counts to the figure legend of Figure 1, as suggested by the reviewer and further adjusted the text to move away from the enumerating style of the text as suggested in the previous point.

2. Sex differentiated effects table is not clear – the effects in males or females columns can be abolished, and the sex with larger effect can be mentioned before each trait-specific p-value. Justification for Multiple testing correction for 97 tests is not sufficient, unless authors state the high correlation between the three studies measures. The legend doesn't make the table content clearer.

Authors' response: Table 2 was adjusted in accordance with the suggestions from Reviewer #1. We have also revised this section to be more comprehensible in accordance with suggestions from Reviewer #2. We hope to have clarified that replicated lead SNPs were tested for sex-heterogenous effects only on the trait that they were associated with. This corresponds to 30 variants that were tested for sex-heterogenous effects on AFR, 44 on LFR and 66 on TFR. Bonferroni correction was used to correct for multiple testing and P-values $< 0.05/140$ (3.57×10^{-4}) were thus considered significant.

3. Line 135 – would suggest mentioning the six loci associated in males combined with the phenotype associated. The authors could speak about sex-specific effects, if association was not significant in one sex. Current terminology is pretty weak.

Authors' response: We have rephrased this section in order to clarify and highlight the loci that were associated with body fat ratios in males. The names of each locus are mentioned as well as the associated trait.

4. Genetic correlation section would benefit from a figure, which would nicely fit as Figure b in current F2.

Authors' response: We have added two illustrations to Figure 2 that concern genetic correlation. Both are correlation matrices. The first one illustrates the correlations between the three body fat ratios within and between sexes. The second illustrates the correlations between the three ratios and anthropometric traits and is analogous to supplementary Table 4. These illustrations have been added as Figure 2b and 2c.

5. Most of the DEPICT results section is enumerating the totals rather than providing biological insights – this part could be improved.

Authors' response: We have adjusted the results section on the enrichment analyses to get away from the enumerating style pointed out by Reviewer #1.

Minor points:

6. The starting phrase of the main text doesn't read well. The phrase "has been shown to be associated with" is heavy and says the same as "is associated".

Authors' response: We have altered the phrase "has been shown to be associated with..." on row 47 to instead say "is associated...".

7. Line 99 is repetitive to line 97.

Authors' response: We have altered this section by removing the first mention of women's larger proportional leg fat mass.

Reviewer #2 (Remarks to the Author):

The manuscript has been substantially shortened and clearer with respect to the methods and presentation of results. There is also effort in providing a global view using enrichment analyses and comparison with other standard anthropometric traits.

1. Tables 1 and 2 are a little disjoint (or I am not reading it correctly). Was GWAMA used on the set of variants taken forth for replication in sex-combined results or the set of 6+71 sex-specific variants that replicated? It is still not clear to me how the sex-heterogenous effects were assessed. Did the authors do a full meta-analysis of the two sets of data? If yes, how did the results differ from the staged approach? I am not fully convinced that a staged approach is more robust and it is messier to assess the different signals at the same locus for males, females. May I suggest that the authors consider a simple flowchart to illustrate all the steps that they have taken in their analyses?

Authors' response: We appreciate the comments on this section from Reviewer #2 and we agree that the staged approach may not be optimal for assessing heterogenous effects between males and females. We have therefore revised the section on GWAMA and the assessment of heterogenous effects between males and females to make it more comprehensible and stringent.

We have removed the section on sex-specific variants that were observed in the sex-stratified GWAS to instead focus on the results from GWAMA. The results from GWAMA should be more robust as this approach includes formal testing for sex-differentiated effects. GWAMA utilizes summary statistics from separate GWAS performed in males and females to test for heterogeneity of allelic effects between males and females.

GWAMA was performed on the replicated trait-associated lead SNPs (n=98). These variants were also only tested for heterogenous allelic effects between males and females in the traits with which they were associated. This corresponds to 30 variants that were tested for sex-heterogenous effects on AFR, 44 on LFR and 66 on TFR. This includes all lead SNPs that replicated. However, several SNPs were associated with more than one trait, which results in a total of 140 tests being performed in the GWAMA analyses. Bonferroni correction was used to correct for multiple testing and P-values $< 0.05/140$ (3.57×10^{-4}) were considered significant. We have revised all parts of the manuscript that are related to this topic and we have reconstructed Table 2 based on our revisions. We have

also constructed a flowchart to illustrate all steps in our analyses in a comprehensible fashion. This is included as Supplementary Figure 6.

3. The authors mentioned that the baseline difference in characteristic of 50,000 individuals in the discovery cohort was unlikely to affect the results from the analyses. Was anything done to assess that? Including study type as covariate or any sensitivity analysis? 50,000 is a third of the discovery.

Authors' response: The study type was adjusted for in our analyses by including the batch variable as a covariate in the analyses. The first release of genotyping data from UK Biobank included the 50,000 individuals that were recruited for the UK Biobank Lung Exome Variant Evaluation (UK BiLEVE). These 50,000 individuals were genotyped on a different array compared to the rest of the participants of UK Biobank, which were genotyped on an axion array with 95% common content as the UK BiLEVE array. We have clarified this in the manuscript in the main text as well as in the methods section.

4. Exact conditional analyses should be used for assessing independence of signals when individual-level data are available. LD is only an approximation and should be used as supporting evidence.

Authors' response: We appreciate the concern by the reviewer that the 'clump' functionality in PLINK may risk missing additional independent signals within associated loci. We therefore conducted exact conditional analyses on all associated loci, for all traits in females and males. This did not reveal any additional signals, which suggests that 'clump' has sufficiently resolved any overlapping signals in our analyses. However, our review of the associated loci revealed a locus that was accidentally omitted from our results, on chr2:408713-466003. This locus has been added to the results and the associated sections of the manuscript have been adjusted to reflect this.

5. It is a little of a stretch to say that missense variants at the loci can be “plausible causal”.

Authors' response: We agree with the reviewer that it may be too optimistic to claim missense variants within the loci to be plausibly causal. We have altered the instance of "Plausible causal missense variants" to just say "Missense variants".

6. “lead SNP” is more commonly used than “leading SNP”.

Authors' response: We have altered all instances of "leading SNP" to "lead SNP" throughout the manuscript.

Reviewer #3 (Remarks to the Author):

The authors addressed many of the issues raised in the previous review of the manuscript. However, issues remain or developed in the revision.

1. A primary claim of the paper is that these analyses identified loci (Table 1) that have not previously been associated with an anthropometric trait, based on LD $r^2 < .1$ with loci listed in the NHGRI-EBI catalog. However, several loci that I checked are not novel using these or even more rigorous criteria. For example, line 203 results text highlights two ‘novel’ loci, KAT8 and ANKDD1B, but at KAT8, rs8050894 reported here is in $r^2 = .86$ (1000Genomes

phase 3 EUR) with rs9925964 reported previously as associated with BMI by Locke et al in 2015; and at ANKDD1B, rs34341 reported here is in $r^2=.54$ with rs6881648 reported as associated with BMI by Akiyama et al in 2017. Further, on line 205, the rs2145270 SNP at highlighted locus BMP2 was itself described previously as associated with BMI by Willer et al 2013. Also, rs2492863 reported here is in $r^2=.99$ with rs2744971 reported as associated with height by Berndt et al in 2013. Other loci may also overlap previous reports. Given the claim that novel biology can be detected by novel loci detected using these AFR, LFR, and TFR traits, the loci need to be reviewed again more rigorously to ensure that they meet the authors' definition of novel. The loci that are truly novel for any anthropometric trait would be interesting and could help explain the basis for differences in body fat distribution.

Authors' response: We apologize for the incorrect designation of novel loci. This appears to have been caused by errors in processing of the GWAS-catalog bulk data file. We have carefully reviewed our designation process and scripts to ensure that the annotation is correct. In addition, we have adjusted all parts of the manuscript that refers to this issue and updated Figure 1 to accurately illustrate novel loci.

2. Based on the further review of loci that are novel for a trait or overlap between traits, Figure 2 likely needs to be updated. The loci that are uniquely associated with the AFR, LFR or TFR traits, not shared by any of the other anthropometric traits, would provide the most novel and interesting information.

Authors' response: We agree with the reviewer that the novel loci are of particular interest. We have therefore updated Figure 2 to highlight novel loci that are not shared with other anthropometric traits. We have also added a subfigure to illustrate the number of loci that overlap between traits.

3. The Table 3 missense variants seem to be mostly from previously reported anthropometric trait loci and do not provide substantial new information. This Table could be moved to the Supplement.

Authors' response: Table 3 has been moved to the supplementary material as supplementary Table 7.

Reviewer #1 (Remarks to the Author):

The authors carefully addressed the reviewers' comments.

In the main text Table 2 with sex-differentiated effects, the effects and their significance in each sex should be reported together with the p-value for heterogeneity.

Unless, I missed something, it would be preferable that effects for all 98 loci (with their assigned [gene] names) are reported for each of three phenotypes in the supplementary table at least.

Currently, it appears that only the effect directions are provided.

It would be important that authors declare to deposit their summary statistics on the web (again, unless I missed it) for the open access to the results. GWAS catalog accepts such depositions.

Reviewer #2 (Remarks to the Author):

The authors have adequately addressed my concerns as well as those from the other two reviewers.

Reviewer #3 (Remarks to the Author):

The authors have addressed my concerns

Response to the Reviewer comments.

Reviewer #1.

1. In the main text Table 2 with sex-differentiated effects, the effects and their significance in each sex should be reported together with the P-value for heterogeneity.

Authors' response: We are unable to include effect sizes and significance for in each sex in Table 2 without violating the formatting recommendations for Nature Communications. We have instead included an additional supplementary data file with the requested data. These data can be found in supplementary Data 2. The other supplementary data files have been reordered accordingly.

2. Unless, I missed something, it would be preferable that effects for all 98 loci (with their assigned [gene] names) are reported for each of three phenotypes in the supplementary table at least. Currently, it appears that only the effect directions are provided.

Authors' response: We agree that the supplementary Figures and Tables document would benefit from a table which includes the effects of representative SNPs at all 98 loci in addition to the material included in Supplementary Data 1. We have included separate tables that include effect estimates and p-values for females and males, as well as the combined cohort in the discovery as well as the replication cohort, for all three phenotypes. These tables are included as supplementary Tables 3-5 in the Supplementary Figures and Tables document, and the supplementary Tables have been reordered accordingly.

3. It would be important that authors declare to deposit their summary statistics on the web (again, unless I missed it) for the open access to the results. GWAS catalog accepts such depositions.

Authors' response: Summary statistics from all association tests are available for download from the file-sharing service at Uppsala University:
https://myfiles.uu.se/ssf/s/readFile/share/3993/1270878243748486898/publicLink/GWAS_summary_stats_ratios.zip.

This information is included in the data availability statement at the end of the methods section. We have also initiated a submission of the data to GWAS Catalog.